# Exploring the Ultrafast Charge-Transfer and Redox Dynamics in Layered Transition Metal Oxides

Guannan Qian, Xiaobiao Huang, Jun-Sik Lee [ID], Piero Pianetta [ID] and Yijin Liu *

Stanford Synchrotron Radiation Lightsource, SLAC National Accelerator Laboratory, Menlo Park, CA 94025, USA
* Correspondence: liuyijin@slac.stanford.edu

**Abstract:** The rapid development and broad deployment of rechargeable batteries have fundamentally transformed modern society by revolutionizing the sectors of consumer electronics, transportation, and grid energy storage. Redox reactions in active battery cathode materials are ubiquitous, complicated, and functionally very important. While a lot of effort has been devoted to investigating redox heterogeneity and its progressive evolution upon prolonged battery cycling, the ultrafast dynamics in these systems are largely unexplored. In this article, we discuss the potential significance of understanding redox dynamics in battery cathodes in the ultrafast time regime. Here, we outline a conceptual experimental design for investigating the ultrafast electron transport in an industry-relevant layered transition metal oxide battery cathode using a plasma-acceleration-based X-ray free-electron laser (FEL) facility. Going beyond the proposed experiment, we provide our perspectives on the use of compact FEL sources for applied research, which, in our view, is an area of tremendous potential.

**Keywords:** free-electron laser; battery; layered cathode

## 1. Introduction

The massive consumption of fossil energy and the associated environmental deterioration have led to global concerns that call for innovations in energy solutions. Energy storage is a vital component in this field because it determines how the energy is stored, transported, and utilized. Lithium-ion batteries (LIBs) are regarded as a transformative energy storage technology, and they have led to broad and profound impacts. High-profile examples of LIB applications include its exponentially growing adoption in consumer electronics and electric vehicles (EVs), both of which have enormous market value [1,2]. In the development of next-generation battery materials that meet the specifications for these commercial applications, there are stringent requirements, e.g., high specific energy, high energy density, stable cycle life, low cost, and improved safety. An in-depth understanding of the fundamental chemical processes and dynamics in LIBs has an important role to play in this quest.

The charging and discharging of a lithium-ion battery involve lithium and electron transport between two electrodes. When the battery is being charged, the externally applied reaction's driving force delithiates the cathode and drives the transportation of the two charge carriers ($Li^+$ and $e^-$) to the anode through different pathways. When the battery is being discharged, the thermodynamically downhill reaction happens spontaneously inside the cell, releasing energy to power the external electric circuits. This is accompanied by redox reactions in the active cathode materials, which, to a great extent, are the ultimate energy reservoirs. Multiple redox-active elements co-exist in the LIB cathode, and they act in a "cooperative" manner to maintain a charge-neutral state at equilibrium. These redox centers demonstrate their respective activities over different voltage windows, could offer different amounts of capacity, and may exhibit different degrees of irreversibility and stability [3]. The charge transfer among different cations/anions, and thus the redox coupling

among different redox centers, could be functionally very important [4]. Collectively, these effects determine several important battery performance metrics, including energy density and cyclability.

Understanding the redox coupling effect in the battery cathode material is imperative. However, some aspects of it remain largely unexplored despite the tremendous investments in this research field. This is due to the fact that the charge transfer among the co-existing redox centers could involve a dynamic process in the ultrafast time regime, engaging multiple redox-active cations and anions. To systematically tackle this problem, X-ray spectroscopic tools with an ultrafast temporal resolution are needed to probe the time-dependent spectroscopic signatures of the transition metal (TM) cations, oxygen anions, and any trace dopant(s) that are present in the cathode. To tackle these complexities, in this article, we discuss our conceptual design for a time-resolved study of battery cathode materials using a plasma-acceleration-based X-ray free-electron laser (FEL). By combining soft X-ray absorption spectroscopy [5] and resonant inelastic X-ray scattering (RIXS) [6] in a laser-pumped X-ray probe experiment, we anticipate that the dynamic interplay among different redox-active cations/anions will be elucidated, offering fundamental insights into the redox coupling effect in the ultrafast time regime. Finally, we provide our perspectives on the use of compact FEL sources for applied research, which, in our view, is an area of tremendous potential.

## 2. X-ray Spectroscopy for Probing the Co-Existing Redox Centers in Battery Cathodes

X-ray spectroscopy has been demonstrated as being a powerful tool for investigating the valence state of various elements of interest in battery cathodes. For example, soft X-ray absorption spectroscopy (sXAS) over the TM *L*-edges and O *K*-edge can probe the valance band of TM 3d and O 2p states [7]. Depending on the detection modality, different sXAS operation modes could offer different insights. Specifically, the Auger electron yield (AEY), total electron yield (TEY), and total fluorescence yield (TFY) all have different probing depths: 1–2 nm for AEY, 2–5 nm for TEY, and 50 nm for TFY, which have been used to reveal the depth-dependent chemical reaction in the battery cathode [8].

Given that multiple co-existing cations and anions in the system could be redox-active under different circumstances, it is important to systematically evaluate their respective redox states and dynamics under different electrochemical conditions. In layered lithium nickel cobalt manganese oxide (NMC), an industry relevant battery cathode material [9], it is the consensus that the Ni cations' redox reaction is responsible for most of the energy storage at a high state of charge [10]. The Ni $L_3$-edge spectrum splits into peaks because of the Ni 2p—Ni 3d electrostatic interaction and the crystal field effects. The ratio between these two peaks is often used to quantify the nickel's oxidation state. This approach has been used to investigate the cathode surface degradation, thermal stability, and the effectiveness of electrode/electrolyte engineering. Similar approaches have also been used to investigate the other prominent TMs in the system, e.g., Co and Mn [11].

In Figure 1, we present our preliminary synchrotron sXAS data over the Ni *L*-edge and O *K*-edge for commercial NMC cathodes at a selected state of charge. The TEY and TFY signals respectively fingerprint the surface and subsurface chemical states for Ni (Figure 1a,b) and O (Figure 1c,d), confirming the chemical sensitivity needed for investigating the TM cation and O anion redox reactions. Moreover, it has also been demonstrated that RIXS over the oxygen *K*-edge can more effectively quantify the oxygen anions' redox activities, which is also reported in our earlier works [12]. These results demonstrate the effectiveness of using X-ray spectroscopy to probe the oxidation states of TM cations and O anions in the cathode materials at equilibrium. To explore the charge transfer and redox dynamics in the ultrafast time regime, however, these spectroscopic tools must be coupled with short-duration X-ray pulses to gain the required temporal resolution.

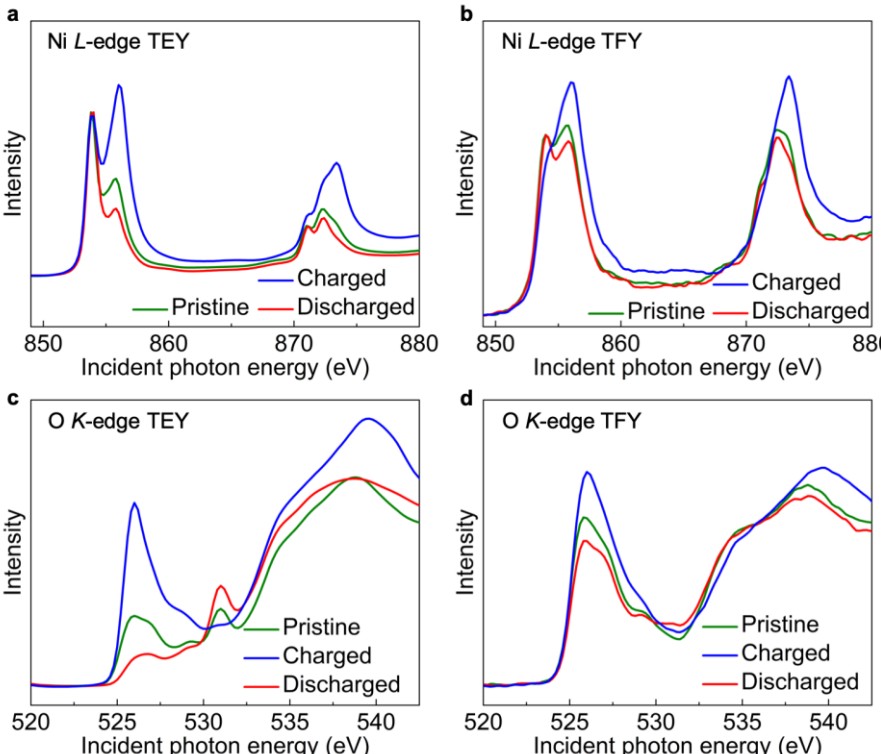

**Figure 1.** The TEY and TFY sXAS spectra over the Ni *L*-edge (**a**,**b**) and O *K*-edge (**c**,**d**) for a selected commercial NMC battery cathode at different states of charge. The apparent differences in the spectral shapes demonstrate sufficient chemical sensitivity for investigating the redox behavior of the elements of interest upon battery charging/discharging.

## 3. The Opportunities Enabled by Plasma-Acceleration-Based FEL

Before further presenting our concept, we discuss the unique technical challenges and opportunities associated with plasma-wakefield-accelerator-based FEL (PWFA-FEL), a concept that was proposed recently [13].

X-ray free-electron lasers (XFELs), such as the Linac Coherent Light Source (LCLS) at SLAC National Accelerator Laboratory [14], provide ultrashort and extremely bright photon beam pulses, which are ideal for probing fast dynamic processes in materials. While the pulse durations for self-amplified spontaneous emission (SASE)- based [15] XFELs are typically of the order of 10 femtoseconds (fs) [14], they can reach few-fs [16] and sub-fs [17,18] levels with special modes of operation. A pulse duration as short as 280 attoseconds (as) through X-ray laser-enhanced attosecond pulse generation (XLEAP) has been reported [19]. Sub-fs X-ray pulses can also be generated with high-harmonic generation (HHG) [20–23], with pulse durations a low as 43 as, as reported in the literature [24]. However, HHG X-ray sources have very low levels of pulse energy (on the order of picojoules (pJ)).

PWFA is a novel type of accelerator that utilizes the plasma wakefield in the "bubble" left behind a driver electron beam after it interacts with a gas medium and expels the electrons in a region around its path [25–28]. Unlike traditional radio-frequency (RF) accelerators, where the acceleration gradient is limited at ~100 MV/m, PWFA can reach an acceleration gradient of ~100 GV/m, enabling a high-energy electron beam over a short distance. As in laser wakefield accelerators (LWFA) [29–31], the driver beam can be replaced with a short, intense laser pulse, which generates a plasma wake in a fashion similar to that of PWFA.

Successful demonstrations of the production of high-energy electron beams in plasma accelerators [28,31] have stimulated widespread interest in utilizing such beams for FELs [32–35]. Compared to FEL-based conventional accelerators, plasma-accelerator-based FELs would have substantially smaller footprints and, consequently, lower construction

costs. This is highly desirable, as it can potentially make FELs widely available at smaller research institutions. However, the beam quality poses tremendous challenges in realizing such FEL facilities.

Electron beams accelerated to high energy through beam-driven and laser-driven plasma wakefield accelerators have similar characteristics. They tend to have a relatively large momentum spread (on the level of a few percent). Additionally, because the accelerating field in the plasma wake varies linearly with the distance behind the driver beam or laser pulse, they acquire a linear energy chirp [36]. These features make it very difficult to utilize PWFA beams for SASE-based FEL since the latter requires electron beams with a small momentum spread (typically $\leq 10^{-3}$).

A novel scheme for producing an FEL with a PWFA was proposed recently [13]. Unlike SASE, which depends on an instability mechanism that arises from the interaction of the electron beam and the photon beam that it emits to generate microbunching on the electron beam on the radiation-wavelength level [15], the new scheme compresses the entire electron bunch down to the radiation-wavelength level. The compression employs the natural energy chirp of the PWFA beams. By sending the strongly chirped PWFA beam bunches through a weak magnetic chicane, the pulse duration can be substantially shortened. Using realistic PWFA-beam parameters, simulations in Ref. [13] demonstrated that the pulse duration can reach sub-100 as, yielding extremely high peak current (near megaampere). Because of the pre-bunching at the radiation-wavelength level, even with a short undulator, the electron beam can produce intense FEL beams. Ref. [13] reports a laser pulse energy of 1.2 mJ at a radiation wavelength of 10 nm, with a pulse duration and peak laser power range of 38-294 FWHM and 0.25–3.8 TW, respectively. It was pointed out that high photon energies can be reached with the same scheme if a high-energy driver beam is used to accelerate the witness beam to a higher energy.

As pointed out in Figure 2 and Ref. [13], the same FEL principle can be applied to laser-driven plasma accelerators. Combining the pre-bunching FEL scheme with laser-plasma accelerators would be more impactful since the required driver laser is more readily available, while the multi-GeV electron driver beam for the PWFA itself needs a large and expensive facility for its production. It can be expected that this combination will cause a paradigm shift in ultrafast XFEL sciences even though it must be left to future studies to explore the full potential of the scheme. Another key component for XFEL experiments is the development of detectors with superior efficiency and robustness. We refer to an article authored by Blaj et al. [37] for a thorough discussion of detector development and damage.

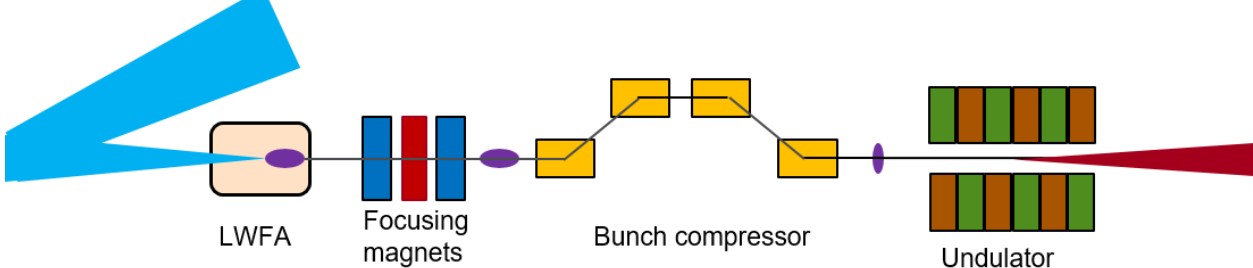

**Figure 2.** Illustration of the plasma-acceleration-based FEL that utilizes the novel pre-bunching scheme proposed in Ref. [13]. Combined with a laser-plasma accelerator, the scheme could enable "table-top" attosecond FELs in the range of VUV to soft X-ray.

## 4. The Hypothesis: Ultrafast Charge Transfer among Different Redox-Active Elements

With the ultrafast temporal resolution facilitated by a plasma-acceleration-based FEL, we now elaborate further on our scientific hypothesis. As we have discussed above, the redox behavior of different TM cations and O anions in an NMC cathode have been extensively studied at or near the state of equilibrium [7]. The ultrafast dynamics in this system, on the other hand, remain largely unexplored. This is due to the fact that the charging

and discharging of a lithium-ion battery involves the physical movement of lithium ions and, therefore, it is considered to be a diffusion-limited process, which is usually not an ultrafast process. However, it is important to point out that electron transport is another indispensable component in this process, which could become a limiting factor in certain circumstances [38]. We conjecture that the electron-transfer induced redox dynamics occur at a level of picoseconds or faster, which necessitates a systematic investigation using ultrafast experimental probes. In Figure 3, we illustrate our hypothesis of a sequential electron transport mechanism in a layered NMC cathode with certain dopant(s) incorporated in its lattice. When a battery is being charged, a lithium ion leaves the cathode lattice, taking away an electron from the surrounding atoms. We know from extensive literature reports that, under normal operational conditions, the lithium deintercalation is eventually associated with the oxidation of the TM cation(s) [10]. However, the electron transport pathway could be more complicated than a direct electron donation from the TM cation to the lithium ion, which is what appears at equilibrium. Intermediate state(s) involving either the oxygen anions or the dopant(s) could exist, as schematically illustrated by the blue and red arrows in Figure 3. As a first step, the dopant ion or the O anion could potentially donate an electron to facilitate the movement of the lithium ion. In the follow-up process, an electron transport between the TM cation and the dopant/O occurs, relaxing the intermediate metastable state. If this is the case, it could largely change our views on the roles of the dopants in the system, which are often simply linked to different material properties if one follows the conventional wisdom [10].

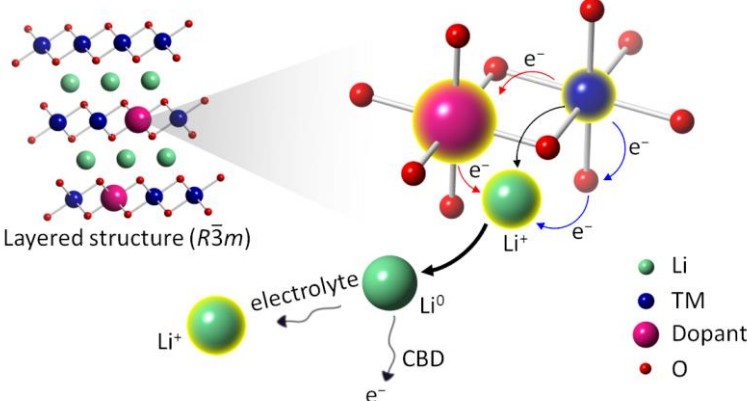

**Figure 3.** The mechanism and the time scale of the dynamic electron transport in a layered transition metal oxide upon lithium intercalation/deintercalation is not fully understood.

Taking a deep dive into this, we conceived of the idea of a laser pump and X-ray (spectroscopy) probe experiment, as illustrated in Figure 4. The NMC cathode can be tuned to different states of charge electrochemically by charging/discharging a standard coin cell to different voltages. The state of the cathode can then be recovered for the pump–probe experiment. A pump laser pulse excites the system by moving an electron above the Fermi level, mimicking an ultrafast charging pulse for the NMC cathode. We can then utilize X-ray spectroscopic probes to monitor the relaxation process for different elements of interest. As discussed above, sXAS over the TM *L*-edges and the O *K*-edges can be used to follow the ultrafast evolution of the respective atoms. We point out here that the critical time scale of this process is not yet clear, and the ultrashort X-ray pulse from a plasma-acceleration-based FEL could be used to survey a wide range of time scales, which will open vast scientific opportunities in this field. For example, there could be important dynamic factors to consider for the selection of dopant(s). This is completely unaccounted for in today's state-of-the-art battery material design. Without an in-depth understanding of the ultrafast charge-transfer and redox dynamics in layered battery cathodes, the dopant selection and optimization are limited to a trial-and-error process, which is inefficient and expensive. Here, we point out that it is important to carefully address the dose management for an

XFEL experiment. In this case, we can compare the equilibrium-state XAS signal before and after different levels of X-ray exposure to determine the maximum tolerable dose and dose rate. It is also useful to note that, using standard battery cells with slow-cycling protocols, we can prepare a sample with a relatively good degree of homogeneity at the electrode scale, which makes it feasible to "refresh" the sample by simply translating the mounted cathode electrode.

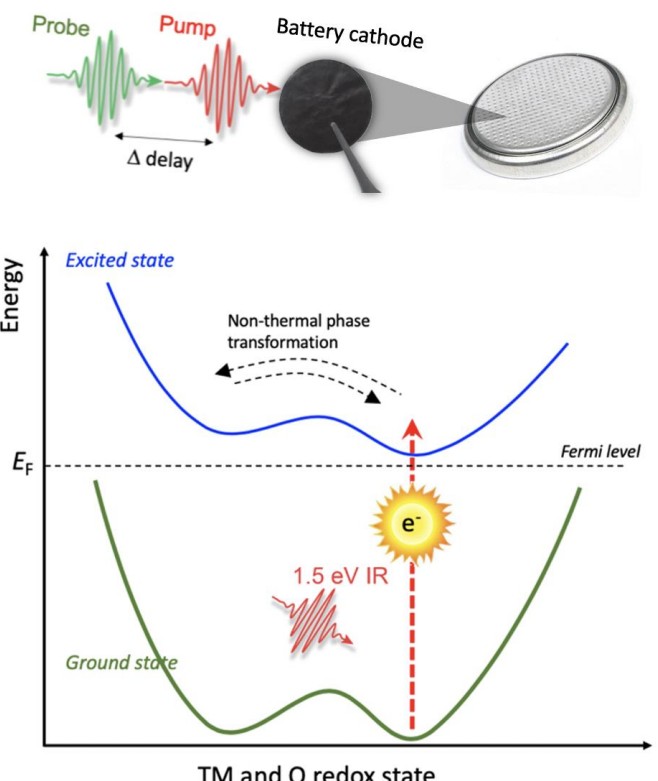

**Figure 4.** A conceptual design for a laser pump and X-ray (spectroscopy) probe experiment on an NMC cathode at different states of charge.

## 5. Perspectives

Battery research is an area of both fundamental interest and practical significance. The challenge in this domain is closely associated with the chemical and physical complexities of real-world batteries. Specifically, the multiscale and hierarchical structure is associated with many different chemical and physical processes, as outlined in Figure 5, in which the horizontal axis is the length scale, the vertical axis is the time scale, and relevant processes are placed into the plot to approximately illustrate their respective time and length scales. It is interesting and important to note that ultrafast dynamics are a largely unexplored territory for battery research. While we acknowledge that the relevance between ultrafast dynamics and battery performance is not very straightforward, in this article, we present a hypothesis-driven design for an experimental concept to elucidate the ultrafast charge-transfer and redox dynamics in layered transition metal oxide battery cathodes. The proposed experiment could elucidate the role of different cations and anions from an unprecedented perspective. It could potentially modulate the design principles for next-generation battery materials.

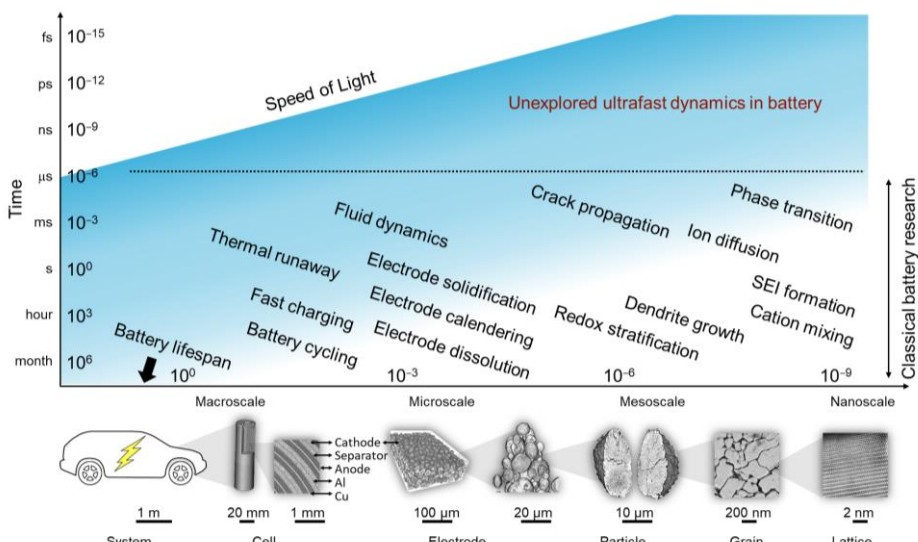

**Figure 5.** Structural, chemical, and dynamic complexities in batteries span a wide range of length and time scales. The ultrafast dynamics are potentially important to fundamental understandings, but they remain largely unexplored. The multiscale structure of a practical battery cell is illustrated in the bottom row. Adapted with permission from [39], Copyright 2021, Elsevier.

The FEL source is indispensable for studying ultrafast dynamics in various material systems. Although large-scale FEL facilities are being built, operated, and upgraded across the world, their accessibility is still very limited. With the development of a small-footprint plasma-accelerator-based FEL with its high-brilliance and ultrashort X-ray pulses, we anticipate that the research of dynamic processes in applied material systems will become a booming area with tremendous potential and impact. It is useful to point out the proposed ultrafast XAS measurements could potentially be complemented using other experimental probes. For example, the ultrafast electron diffraction method could be used to monitor the dynamics of lattice distortion and relaxation, which could affect the local resistance of ion transport. An ultrafast optical Raman probe could be used to fingerprint the local state-of-charge evolution if the Raman signals are calibrated carefully. When applied in the ultrafast time regime, a multimodal experimental approach could open vast scientific opportunities.

**Author Contributions:** Conceptualization, Y.L.; experiment, G.Q. and J.-S.L.; investigation, all authors; writing—original draft preparation, G.Q., X.H. and Y.L.; writing—review and editing, J.-S.L. and P.P. All authors have read and agreed to the published version of the manuscript.

**Funding:** Stanford Synchrotron Radiation Lightsource, SLAC National Accelerator Laboratory, is supported by the U.S. Department of Energy (DOE), Office of Science, Office of Basic Energy Sciences, under Contract No. DE-AC02-76SF00515.

**Data Availability Statement:** All data that support the findings of this study are available from the corresponding authors upon reasonable request.

**Acknowledgments:** The authors gratefully thank Jagjit Nanda of SLAC and Hoyoung Jang and Soonnam Kwon of PAL-XFEL for valuable discussions.

**Conflicts of Interest:** The authors declare no conflict of interest.

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
