# Peer review of "Exploring the Ultrafast Charge-Transfer and Redox Dynamics in Layered Transition Metal Oxides"

_condensedmatter, doi:10.3390/condmat8010025_

Round 1

Reviewer 1 Report

In this work, the authors discuss the significance of understanding the redox dynamics in battery cathode in the ultrafast time regime. And the authors propose a conceptual experimental design for investigating the ultrafast electron transport in layered transition metal oxide battery cathodes by using plasma acceleration-based X-ray free electron laser (FEL) facility. Perspectives on the use of compact FEL sources for applied research have been provided. These new insights can serve as design principles for future discovery of redox dynamics for LIBs.

 In regard to this manuscript, the reviewer raises the following questions:

1.       In the introduction section, the authors should be clear about the time scale for redox process of cations and anions.

2.       In line 88, “oxygen cations redox activities” or “oxygen anions”?

3.       It is not clear in the discussion section why this ultrafast redox coupling process has any significant impact on the battery overall performance, like cycling, etc.

4.       The authors should discuss about the detector for ultrafast measurement.

5.     The authors need to cover the ultrafast measurement effort based on other techniques to compensate for the X-ray probes.

Reviewer 2 Report

*** Referee report ***

Manuscript condensedmatter-2246395

Guannan Qian et al. "Exploring the ultrafast charge transfer and redox dynamics in layered transition metal oxides"

The authors present an interesting experimental soft X-ray FEL-based method 

to monitor reduction dynamics in battery cathode materials with sub-ps time resolution.

I totally agree with the authors that time resolved XAS is a really 

promising experimental technique for investigating e developing 

ion batteries. TR-XAS is very sensitive to charge transfer in condensed matter

and L-edge XAS is suitable to monitor unoccupied valence d-states 

across the Fermi level in transition metals, thus representing 

the ideal scheme to study ultrafast electron dynamics in cathode materials.

Availability of FEL tr-XAS with future plasma acceleration sources

is undoubtedly a unique opportunity for developing time resolved methods

for ion batteries studies.

The limit of the manuscript is that presented XAS spectra have been carried out 

at synchrotrons. The signal-to-noise ratio in FEL-based tr-XAS experiments

is drastically worse. The authors should discuss this point in the manuscript,

describing also X-ray FEL experimental setups and methodologies.

Moreover, the authors mention the high photon intensity attainable by FEL sources,

emphasizing in contrast the poor intensity of HHG sources. This point is rather critical,

as high FEL pulse intensities likely damage the samples. On the other hand 

decreasing the intensity of the probe affects the signal-to-noise ratio.

The authors should clarify if their envisaged tr-XAS experiments 

should be carried out in a single shot fashion (high intensity)

or with a prolonged exposure (low intensity).   

The intensity level of the laser pump should be clarified too as 

the predicted reduction dynamics could be not measurable by tr-XAS 

if the pump is not intense enough. 

Furthermore, the authors should address the issue of the limited

attenuation length of visible light in transition metals (about 10 nm) 

What is the thickness of the cathode specimen under examination?

Can the laser pump trigger the ultrafast cathode reduction 

in the entire sample?

Mentioning the attosecond time scale, possibly achievable with PWFA-based FELs,

appears misleading for tr-XAS experiments as the collective charge transfer 

driven by a fs-laser pulse is expected to alter the XAS spectra 

in the fs time scale. The authors should better clarify this concept. 

Is attosecond time resolution really needed for the proposed experiments?

Finally, I would like that authors better explain why the proposed 

laser-driven electron dynamics in tr-XAS experiments should be effective for mimicking 

the real cathode reduction occurring in ion batteries.

In conclusion, I recommend to publish the manuscript after some revisions

as suggested in this report.

Reviewer 3 Report

The manuscript aims to propose a new method for investigating the ultrafast charge transfer and redox dynamics in layered transition metal oxides in common devices such as a coin cell, using X-ray free electron laser and sXAS spectra. It is well-written and well-organized, providing a clear and concise description of the research project's objectives, methodologies, and potential impacts. The research project proposed in this proposal is highly innovative and has significant potential to advance our understanding of the charge dynamics in battery cathode on the ultrafast time scales. The proposed research utilizes cutting-edge techniques such as ultrashort X-ray free electron laser and sXAS spectra, which are crucial in providing insights into the ultrafast dynamics of electronic and chemical processes. I believe that the proposed research project has significant potential to make significant contributions to the field of ultrafast spectroscopy. Therefore, I generally recommend the publication of this manuscript. However, I suggest that the authors provide more detailed information on addressing my following questions.

1. How strong does pump pulse need to be to induce the ultrafast charging process like the process shown in Figure 1? Will the pump pulse generate other non-thermal effects such as the ultrafast dilation of the lattice interfere the desired phenomenon?

 2. How does this method directly detect the dopants charge states if the authors aim at discovering ultrafast dopant-mediated charge transfer?
